# Development of a Sensitive Digital Droplet PCR Screening Assay for the Detection of *GPR126* Non-Coding Mutations in Bladder Cancer Urine Liquid Biopsies

**DOI:** 10.3390/biomedicines11020495

**Published:** 2023-02-08

**Authors:** Mark Jain, Alexander Tivtikyan, David Kamalov, Savva Avdonin, Tagir Rakhmatullin, Eduard Pisarev, Maria Zvereva, Larisa Samokhodskaya, Armais Kamalov

**Affiliations:** 1Medical Research and Educational Center, Lomonosov Moscow State University, 119992 Moscow, Russia; 2Department of Fundamental Medicine, Lomonosov Moscow State University, 119991 Moscow, Russia; 3Department of Bioinformatics and Bioengineering, Lomonosov Moscow State University, 119991 Moscow, Russia; 4Department of Chemistry, Lomonosov Moscow State University, 119991 Moscow, Russia

**Keywords:** liquid biopsy, urine, bladder cancer, digital droplet polymerase chain reaction, *TERT* promoter, *GPR126*, cell-free DNA, non-coding mutations

## Abstract

Recent whole-genome sequencing studies identified two novel recurrent mutations in the enhancer region of *GPR126* in urothelial bladder cancer (UBC) tumor samples. This mutational hotspot is the second most common after the *TERT* promoter in UBC. The aim of the study was to develop a digital droplet PCR screening assay for the simultaneous detection of *GPR126* mutations in a single tube. Its performance combined with *TERT* promoter mutation analysis was evaluated in urine of healthy volunteers (*n* = 50) and patients with cystitis (*n* = 22) and UBC (*n* = 70). The developed assay was validated using DNA constructs carrying the studied variants. None of the mutations were detected in control and cystitis group samples. *GPR126* mutations were observed in the urine of 25/70 UBC patients (area under the ROC curve (AUC) of 0.679; mutant allele fraction (MAF) of 21.61 [8.30–44.52] %); *TERT* mutations–in 40/70 (AUC of 0.786; MAF = 28.29 [19.03–38.08] %); ≥1 mutation–in 47/70 (AUC of 0.836)). The simultaneous presence of *GPR126* and *TERT* mutations was observed in 18/70 cases, with no difference in MAFs for the paired samples (31.96 [14.78–47.49] % vs. 27.13 [17.00–37.62] %, *p* = 0.349, respectively). The combined analysis of these common non-coding mutations in urine allows the sensitive and non-invasive detection of UBC.

## 1. Introduction

Due to both advances in the molecular characterization of tumor cells and the development of sensitive analytical techniques, liquid biopsy began to gradually replace traditional invasive methods for cancer diagnosis and monitoring [1]. The term “liquid biopsy” is used for various approaches for non-invasive detection of tumor-derived targets, such as cell-free tumor DNA (tDNA), exosomes, circulating tumor cells, tumor-educated platelets, etc., in bodily fluids [2,3,4,5,6]. In the case of urothelial bladder cancer (UBC), the most suitable liquid substrate for the analysis is, obviously, the urine, as it is in a continuous contact with the tumor surface. This biomaterial contains not only cancerous cells shed from the bladder wall, but also cell-free tDNA, which is secreted by the tumor either actively or as a result of cell death [7].

Mainly, cell-free tDNA is distinguished from wild type cell-free DNA molecules via the detection of certain hotspot mutations, which are known to occur only in cancer cells [5,8]. However, other approaches are proposed, such as the analysis of epigenetic markers and cell-free DNA fragmentation patterns [9,10].

The advent of large cancer-related metagenomic projects (“Catalogue of Somatic Mutations in Cancer” and “The Cancer Genome Atlas Program”, among others) substantially simplified the selection of optimal mutational hotspots, especially in coding regions of the genome [11,12]. Nevertheless, according to these data, it was thought initially that UBC has a quite diverse mutational signature with no universally common hotspots, as the three most prevalent missense mutations in UBC (*FGFR3* S249C, *PIK3CA* E545K, and *ERBB2* S310F) have a fairly low incidence rate–7.1%, 6.6%, and 4.7%, respectively [12].

Therefore, detection of UBC based on urine DNA analysis with sufficiently high sensitivity was possible only using next-generation sequencing (NGS) technologies, whereas the application of approaches that are more targeted and applicable in clinical medicine, such as various types of polymerase chain reaction (PCR), was suboptimal. However, the outlook on this problem changed in 2013, when exceptionally frequent (59–77%) *TERT* promoter mutations were identified in UBC [13,14,15]. Subsequently, researchers world-wide successfully demonstrated that the detection of these mutations in urinary DNA using digital droplet PCR (ddPCR) enables UBC diagnosis with both high sensitivity (46.0–67.4%) and specificity (100%) [16,17,18].

Recently, a new breakthrough in the study of UBC genetics happened–whole-genome sequencing revealed highly recurrent mutations in the sixth intron of *GPR126* (also known as *ADGRG6*), namely chr. 6: 142,706,206, G/A substitution and chr. 6: 142,706,209, C/T substitution (assembly GRCh37) [19]. Shortly after, these results were confirmed in large cohorts of patients with an overall mutation frequency of 31.4–53.0%, defining the sixth intron of *GPR126* as the second most common mutational hotspot in UBC after the *TERT* promoter [20,21,22]. In the structure of *GPR126* non-coding mutations the G/A substitution is the most frequent (~57%), followed by the C/T substitution (~26%) and by the simultaneous presence of G/A and C/T substitutions (~17%) [20]. The relatively high frequency of all three genotypes (as well as the fact that mutational sites are spaced only by two nucleotides) is a challenge for assay design, as ddPCR-based analysis would require three sets of detection probes for each possible mutant genotype (and three corresponding separate reactions).

The aim of the present study was to develop a screening assay which would enable the simultaneous detection of all three mutated *GPR126* genotypes in a single tube using ddPCR as an analytical method of choice. Additionally, the performance of this assay combined with *TERT* promoter mutation analysis was evaluated in urine liquid biopsy samples of UBC patients.

## 2. Materials and Methods

### 2.1. General Information

The study was approved by the institution’s Local Ethics Committee (#4/20, 27 April 2020) and conducted according to the tenets of the Declaration of Helsinki. Patients were enrolled from April 2020 to February 2022. All participants provided signed informed consent forms. The study included 70 patients (of which 27 took part in our previously published pilot study dedicated to the analysis of *TERT* promoter mutations [17]) with histologically confirmed non-muscle-invasive UBC (UBC group), 22 patients with various types of cystitis (cystitis group), and 50 volunteers (control group) without any known urological malignancies, without a family history of UBC, and without UBC risk factors such as tobacco smoking and occupational exposure to carcinogens (e.g., various aromatic amines, which are used in the rubber and dye industries [23]). Tumor size parameters were evaluated using transabdominal 3D ultrasonography on a SonoAce X8 instrument (Samsung Medison Co., Ltd., Seoul, Republic of Korea). The demographic and clinical characteristics of study participants are presented in Table 1.

### 2.2. Biomaterial Collection and Processing

All study participants donated ~50 mL of first morning urine. In the UBC and cystitis groups, biomaterial was obtained prior to any diagnostic/surgical interventions (therefore, the diagnosis was verified shortly after the collection of biomaterial). Intact urine samples were immediately stored at −80 °C. After defrosting, urine samples were thoroughly mixed by pulse-vortexing. DNA was isolated from all UBC and cystitis groups and 25 control group samples using QIAamp Circulating Nucleic Acid Kit (Qiagen GmbH, Hilden, Germany) according to the instruction manual (urine input volume—4 mL). However, the lysis stage was extended for an additional 20 min, as the standard manufacturer’s protocol was designed for urine supernatant and not for intact biomaterial, which could contain substantially more cell fragments. DNA isolation from the remaining 25 control group samples was conducted using Quick-DNA Urine Kit (Zymo Research Corp., Irvine, CA, USA), and urine input volume was 15 mL. Note that samples processed using this kit were stored with an appropriate volume of Urine Conditioning Buffer provided by the manufacturer. In all cases DNA elution volume was set to 100 µL.

The use of a different extraction protocol with a substantially increased urine input volume for half of the control group samples was intentional. This was carried out to replicate the total urine DNA concentration in samples from the UBC group with macrohematuria and/or abundance of nucleated cells. Values of wild type DNA in all analyzed urine samples with indication of the nucleic acid extraction kit used are presented in Appendix A.

### 2.3. DNA Analysis

All experiments were carried out using the QX200 AutoDG ddPCR System, and the amplification was performed on the CFX96 Touch instrument (Bio-Rad Laboratories, Inc., Hercules, CA, USA). Droplet generation, amplification, and droplet analysis were conducted according to the manufacturer’s instructions.

The developed *GPR126* non-coding mutation screening assay included one pair of primers, one HEX-labeled wild type probe, and three FAM-labeled mutation probes (for chr. 6: 142,706,206, G/A substitution; chr. 6: 142,706,209, C/T substitution; and for the simultaneous presence of both substitutions, assembly GRCh37). The composition of the assay is schematically presented in Figure 1. Oligonucleotide sequences as well as their intended concentrations in the final ddPCR mixture are available in Table 2. The amplicon length is 121 bp. The following amplification protocol was used: incubation at 95 °C (10 min); 41 cycles of denaturation at 94 °C (30 s) and annealing/extension at 53.7 °C (1 min); and incubation at 98 °C (10 min). The cutoff value for the ddPCR false-positive signal was evaluated based on the analysis of the control group urinary DNA samples and subtracted from the tDNA analysis results.

Note that this screening assay can detect the presence of either of the above-mentioned mutations in a single tube, but it is not suitable for their discrimination between each other, as all mutation probes are labeled with the same fluorophore. However, if necessary, the screening assay can be converted to a regular one. For this purpose, only one select mutation probe could be added to the assay (instead of all three at the same time), and its concentration should be tripled.

The ddPCR analysis of the *TERT* promoter mutations at two positions upstream of the transcription starting site (at −124 bp, chr. 5: 1,295,228, G/A substitution; and −146 bp, chr. 5: 1,295,250, G/A substitution, assembly GRCh37) was carried out using TaqMan Liquid Biopsy dPCR Assays Hs000000092_rm and Hs000000093_rm (Thermo Fisher Scientific, Inc., Waltham, MA, USA), respectively. The thermocycling conditions were set according to the manufacturer’s protocol. The false-positive signal cutoff value was established as previously described and subtracted from the tDNA analysis results as well [18].

### 2.4. Synthetic DNA Constructs

To validate the developed ddPCR screening assay, three synthetic DNA constructs, each carrying one of the three possible *GPR126* non-coding mutations (chr. 6: 142,706,206, G/A substitution; chr. 6: 142,706,209, C/T substitution; and simultaneously both substitutions, assembly GRCh37), were designed. These DNA constructs were created by Evrogen JSC. In brief, a fragment of the *GPR126* sixth intron (chr. 6: 142,705,865–142,706,619, assembly GRCh37) was amplified and purified from Raji cell line DNA (Evrogen JSC, Moscow, Russia). The PCR product was cloned using pAL2-T vector (Evrogen JSC, Moscow, Russia). The site-directed mutagenesis was used to introduce the mutations and create three corresponding DNA constructs carrying the *GRP126* sixth intron: G/A, C/T, and G/A + C/T substitutions. Sanger sequencing assembly data for all three mutated DNA constructs as well as for the wild type clone are available in Appendix A.

### 2.5. Statistical Analysis

Data were analyzed using IBM SPSS Statistics 26.0 Software (IBM Corp., Armonk, NY, USA). Results of quantitative tDNA analysis are presented as mutant allele fraction (MAF):MAF = C_mutated DNA_/(C_wild type DNA_ + C_mutated DNA_)
where «C» is the DNA concentration expressed in copies per microliter of final ddPCR reaction mixture.

Data distribution normality was evaluated using Shapiro–Wilk’s test. Due to the absence of normal distribution for all tested variables, non-parametrical statistical tests were used. Paired data were compared using Wilcoxon test, whereas unpaired data–using Mann–Whitney U test. Spearman’s rank correlation coefficient (r_s_) was used to assess associations between two variables. Coefficient of determination (R^2^) was calculated to test the linearity of the MAF analysis. Fisher’s exact test was applied to compare qualitative data. Quantitative data are presented as median [interquartile range]. Receiver operating characteristic (ROC) analysis was carried out to evaluate the diagnostic potential of the assays and to determine false-positive signal cutoff value; *p* < 0.05 was considered statistically significant.

## 3. Results

### 3.1. Assay Validation

The developed ddPCR screening assay allowed the successful detection all three synthetic DNA constructs carrying *GPR126* non-coding mutations which were spiked in wild type DNA samples. Examples of ddPCR two-dimensional scatter plots are presented in Figure 2.

To determine the linearity of the quantitative analysis, a series of wild type DNA samples spiked with mutation-carrying DNA constructs at a broad range of expected MAFs (0.5–50%) were subjected to the analysis. The developed assay exhibited high linearity with R^2^ coefficients of 0.923, 0.986, and 0.902 for the analysis of *GPR126* sixth intron G/A, C/T, and G/A + C/T substitutions, respectively (Figure 3).

As is shown in Figure 2, occasionally false-positive droplets with *GPR126*-mutated DNA appeared in wild type samples. Analysis of the control group urinary DNA revealed that the concentration of false-positive mutant alleles (expressed in copies per microliter of reaction mixture) in these samples correlates with the overall DNA input (r_S_ = 0.593, *p* < 0.001). Thus, the false-positive threshold should be set for the MAF, rather than for the absolute DNA concentration itself. The false-positive MAF in the control group was 0.31 [0.15–1.13] %, with a maximum observed value of 1.79%. As theoretically *GPR126* hotspot mutations should not be present in the urine of healthy donors, the cutoff value calculated during the ROC analysis (1.85%) was set at a specificity of 100%, and this value was further subtracted from the MAFs of UBC group samples to determine their true MAF values. To prove that the false-positive droplets are in fact analysis artefacts, nine samples in the control group and nine mutation-positive samples in the UBC group were randomly selected to be analyzed in replicates. Then, the standard deviation of the MAF expressed in percent of the mean value was calculated. It appeared that the false-positive MAF in the control group varied significantly higher than the true MAF in the UBC group (31.4 [15.7–59.5] vs. 8.2 [3.9–19.1], *p* < 0.05, respectively), indicating the random nature of the false-positive signal in wild type DNA.

### 3.2. Urine Liquid Biopsy Analysis

After the implementation of optimal false-positive signal cutoff values, *GPR126* and *TERT* non-coding mutation ddPCR analysis demonstrated the complete absence of tDNA in the control and cystitis groups samples. ROC curves for the analysis of both mutational hotspots as well as for their combination are presented in Figure 4a. *GPR126* sixth intron mutations were detected in the urine of 25/70 UBC patients (area under the ROC curve (AUC) of 0.679; MAF = 21.61 [8.30–44.52] %), whereas *TERT* promoter mutations–in 40/70 cases (AUC of 0.786; MAF = 28.29 [19.03–38.08] %). *TERT* promoter G/A substitution at −124 bp was substantially more frequent than G/A substitution at −146 bp (37/70 vs. 10/70). At least one of the analyzed mutations was detected in 47/70 UBC urine samples (combined AUC of 0.836).

The simultaneous presence of *GPR126* and *TERT* non-coding mutations was observed in 18/70 samples. As shown in Figure 4b, there was no statistically significant difference for MAFs in the paired samples (31.96 [14.78–47.49] % vs. 27.13 [17.00–37.62] %, *p* = 0.349, respectively).

No correlation was found between tumor size and MAF determined for either of the studied mutations or for their combination (*p* > 0.05) (Appendix A). The absence of significant associations was observed between any qualitative/quantitative urine tDNA analysis data and UBC grade, stage, history of smoking, and age (*p* > 0.05), except moderate correlation for *GPR126* MAF and age (r_S_ = 0.481, *p* = 0.015), although not for its detection rate (*p* > 0.05) (Appendix A). Finally, as the developed *GPR126* non-coding mutation screening assay had a relatively high false-positive MAF threshold, the impact of macrohematuria was considered. This condition may lead to the abundant presence of genomic DNA from leucocytes in urine and, thus, to an artificial decrease in MAF. However, in the present cohort, urine samples with macrohematuria did not statistically significantly differ from the rest (30.78 [17.63–42.33] % vs. 21.12 [4.45–46.56] %, *p* = 0.543) (Appendix A).

## 4. Discussion

Bladder cancer is known for its exceptionally high recurrence rate (up to 74.3% within 10 years) [24]. The necessity of a regular (3–6 months) cystoscopy and urine cytology follow-up for a period of 5 years, and then yearly after tumor resection, makes the patient management in this disease to be one of the most expensive among all cancers [25,26]. Therefore, the introduction of sensitive analytical approaches for urine liquid biopsy analysis into clinical practice could drastically decrease the costs.

The screening assay proposed in the present study enabled simultaneous detection of three possible substitutions in the *GPR126* mutational hotspot in a single tube, substantially reducing the consumption of ddPCR reagents and DNA samples, which in turn allows to isolate urinary DNA in a lower volume, thus, at a higher concentration. The detection rates of the studied mutations in urine were close to the expected rates based on the reported frequencies of these genetic alterations in tumor tissue (35.7% vs. 31.4–53.0% for the *GPR126* sixth intron and 57.1% vs. 59.0–77.0% for the *TERT* promoter, respectively) [13,14,15,20,21,22]. Overall, tDNA was detected in 67,1% of the UBC group samples, indicating that although these two most common UBC mutational hotspots achieved substantial tumor genotype coverage, an extension of the panel is still required.

In concordance with our results, the diagnostic sensitivity of the exclusive analysis of *TERT* promoter mutations in urinary DNA using ddPCR varies in the available publications from 46.4% to 68.3% [16,17,18,27,28]. In a recent study, Hayashi et al. demonstrated that the inclusion of *FGFR3* S249C mutation analysis increases the sensitivity of this approach an additional 8% [29]. Although ddPCR has certain advantages over NGS, such as relatively cheap and rapid sample processing and independence of bioinformatics data interpretation, its major drawback is the limitation in the number of analytical targets per DNA sample [30]. Urine liquid biopsy approaches based on NGS already achieve a sensitivity of 84.0–97.4% and specificity of 97.0–100.0%, and these panels include up to 50 recurrently mutated in UBC genes, such as *TERT*, *FGFR3*, *TP53*, *PIK3CA*, *ERBB2*, *KRAS*, and others [31,32,33]. Despite limited effectiveness in initial diagnosis and screening, ddPCR may truly excel in the field of UBC post-treatment surveillance [34]. In this case, resected material may be sequenced, therefore a single personalized ddPCR mutation assay is needed to monitor the recurrence of the disease, as it was recently demonstrated by Pritchard et al. [35]. A plethora of novel non-coding mutations in UBC was recently discovered (hotspots in *PLEKHS1* (38,7%), *TBC1D12* (25.5%), *LEPROTL1* (23.8%), and *WDR74* (17.2%)) [20]. Even though alterations in these hotspots are not as common as in *GPR126*, their addition to the urine liquid biopsy mutation panel could substantially increase its sensitivity, still being in the range of the analytical target quantity, where ddPCR is preferred over sequencing approaches.

*GPR126* encodes a G protein-coupled receptor which is known to bind collagen IV and laminin-211, mediating the myelination process [36]. It was demonstrated that the sixth intron of this gene comprises its enhancer, and that G/A and C/T substitutions (chr. 6: 142,706,206 and 142,706,209, respectively, assembly GRCh37) reduce the expression of *GPR126*, which in turn compromises the ability of UBC cells to recruit endothelial cells [19]. However, the exact role of these mutations in UBC tumorigenesis is yet to be uncovered. On the other hand, the impact of *TERT* promoter mutations is less elusive, as it is known that they promote its transcription, leading to elevated telomerase activity and, thus, limitless cell proliferation driven by oncogenes [37]. Overall, non-coding mutations in both genes are considered to be early events in tumor formation, and this is supported by the fact that in the present study samples positive for both hotspots exhibited absences of statistically significant differences in the pairwise MAF comparison, as well as by the observed equal distribution of mutations across various tumor stages. Moreover, recently, it was demonstrated that *TERT* promoter mutations are detectable up to 10 years prior to clinical manifestation of UBC [38].

The observed absence of significant correlation between tumor size and MAF in the present study was generally expected. As it was discussed previously [18], there are several factors impeding the estimation of tumor size at initial diagnosis based on urinary tDNA level. Firstly, visualization techniques used to evaluate tumor parameters have certain limitations, mainly intra- and interobserver variability [39]. Secondly, MAF could be distorted in samples with the abundant presence of genomic DNA (leukocyturia/active cell shedding), whereas absolute tDNA values are heavily influenced by the DNA isolation efficacy. Lastly, even though analyzed mutations probably occur in the early stages of tumorigenesis, it is not guaranteed that ~100% cells of the observed at instrumental analysis neoplasm volume really have these genetic alterations.

This study had certain limitations. The control group consisted of healthy volunteers with no signs of urological disease, whereas the cystitis group was rather small. In clinical application, the analysis would primarily have been carried out in differential diagnosis of patients with hematuria. Mutational status of the urinary DNA was not compared to that of the paired tumor DNA samples, which may be useful to verify the absence of false mutation calls. However, the robust determination of false-positive signal cutoff values (resulting in 100% specificity) at least to some extent eliminates the need to evaluate the concordance between samples. As the presented ddPCR screening assay did not allow to discriminate between *GPR126* sixth intron mutations, the incidence of individual substitutions and their relation to pathological characteristics of tumors was not studied.

## 5. Conclusions

Urine liquid biopsy is a promising non-invasive tool for the diagnosis and management of UBC. Combined analysis of non-coding mutations in *GPR126* and *TERT* genes in the present study allowed to detect tDNA in the majority of UBC samples. Owing to its exceptionally high prevalence in this disease, a mutational hotspot in the *GPR126* gene must become a new staple in diagnostic panels for the analysis of urinary tDNA. Moreover, being an early genetic event in the tumorigenesis, non-coding mutations in *GPR126* may be an appealing target for the post-treatment monitoring of UBC recurrence. Further studies not only evaluating the diagnostic efficacy of urine liquid biopsy but also assessing the potential economic benefit of the translation of this approach into clinical practice are awaited.

## Figures and Tables

**Figure 1 biomedicines-11-00495-f001:**
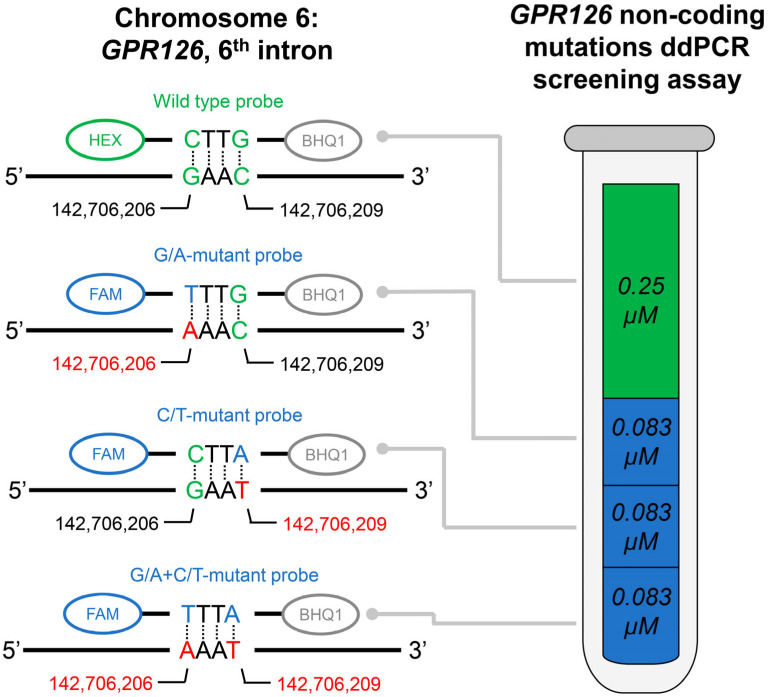
Schematic description of the developed *GPR126* non-coding mutations ddPCR screening assay. Actual fluorescent probes used in the assay are designed to be complementary to the antisense DNA strand; complete oligonucleotide sequences are available in Table 2. Probes were dual-labeled with Hexachlorofluorescein (HEX)/6-Carboxyfluorescein (FAM) fluorescent dyes and Black Hole Quencher 1 (BHQ1) non-fluorescent chromophore. The concentrations of the probes are indicated for the final ddPCR reaction mixture. Positions of the nucleotides in the genome are presented according to the assembly GRCh37.

**Figure 2 biomedicines-11-00495-f002:**
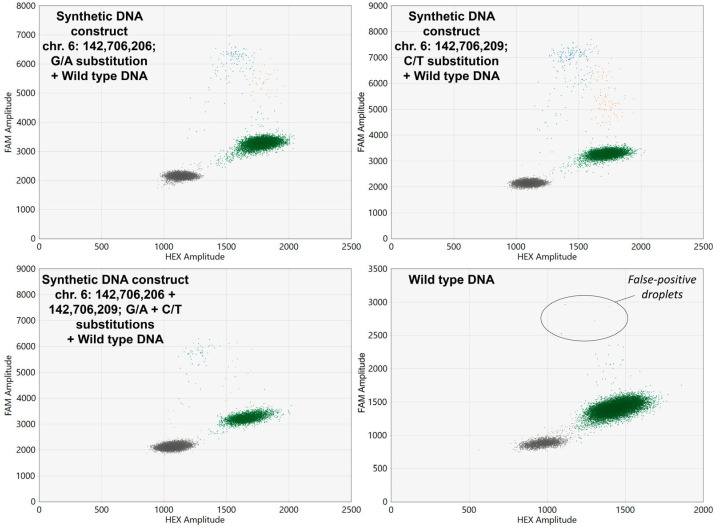
Two-dimensional ddPCR scatter plots for the analysis of wild type DNA spiked with synthetic DNA constructs carrying *GPR126* non-coding mutations. HEX, Hexachlorofluorescein; FAM, 6-Carboxyfluorescein. Wild type probes are HEX-labeled, whereas mutant probes are FAM-labeled. Colored dots represent single droplets of emulsion carrying amplified mutated DNA (blue), wild type DNA (green), and mutated + wild type DNA (orange). Positions of the nucleotides in the genome are presented according to the assembly GRCh37.

**Figure 3 biomedicines-11-00495-f003:**
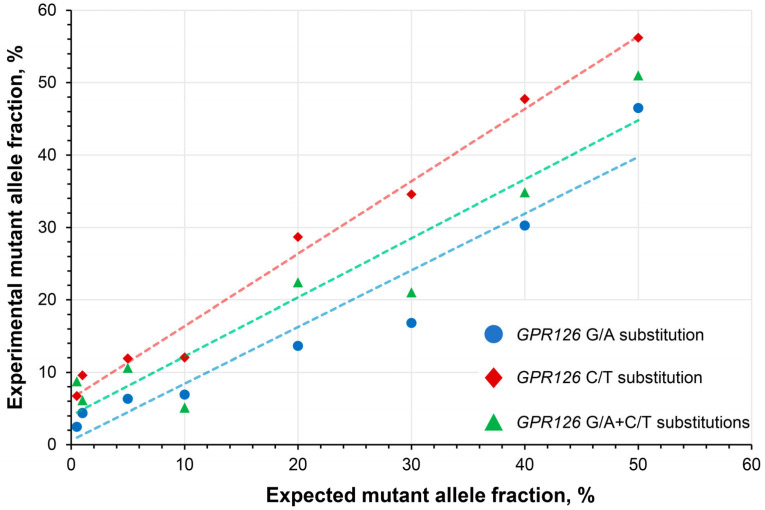
Assessment of the quantitative analysis linearity in wild type DNA spiked with synthetic DNA constructs carrying GPR126 non-coding mutations at different expected mutant allele fractions. *GPR126* sixth intron G/A substitution–chr. 6: 142,706,206, C/T substitution–chr. 6: 142,706,209 (assembly GRCh37).

**Figure 4 biomedicines-11-00495-f004:**
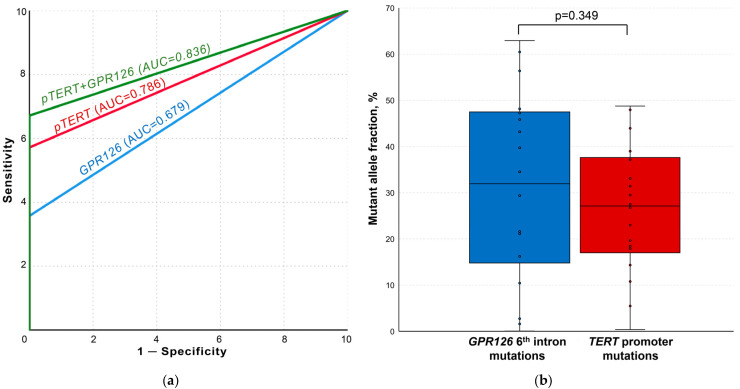
Results of the urine liquid biopsy analysis. (**a**) ROC curves for *TERT* and *GPR126* non-coding mutation analysis. (**b**) Comparison of paired samples simultaneously containing both *GPR126* and *TERT* non-coding mutations. AUC, area under the curve; *pTERT*, *TERT* promoter. False-positive signal cutoff values for all the implemented assays were subtracted from the results of each sample prior to the analysis.

**Table 1 biomedicines-11-00495-t001:** The demographic and clinical characteristics of study participants.

Parameters	UBC Group	Cystitis Group	Control Group
	(*n* = 70)	(*n* = 22)	(*n* = 50)
Age, years ^1^	63 (26–87)	52 (27–66)	20 (18–41)
Sex, *n* (%):			
- Male	61 (87.2%)	12 (54.5%)	24 (48%)
- Female	9 (12.8%)	10 (45.5%)	26 (52%)
Smoking, *n* (%)	23 (32.9%)	2 (9.1%)	0 (0%)
Urinary retention, *n* (%)	16 (22.9%)	8 (36.4%)	0 (0%)
Macrohematuria, *n* (%)	7 (10%)	6 (27.3%)	0 (0%)
Tumor stage, *n* (%):			
- I	60 (85.7%)	N/A	N/A
- II	2 (2.9%)	N/A	N/A
- IIIA	6 (8.6%)	N/A	N/A
- IIIB	1 (1.4%)	N/A	N/A
- IVA	1 (1.4%)	N/A	N/A
Tumor grade, *n* (%):			
- Low	33 (47.1%)	N/A	N/A
- High	20 (28.6%)	N/A	N/A
- Unknown	17 (24.3%)	N/A	N/A
Tumor size, cm^3 2^	2.16 [1.04–11.05]	N/A	N/A
Metastasis, *n* (%)	1 (1.4%)	N/A	N/A

UBC, urothelial bladder cancer; N/A, not available; ^1^ data presented as mean (range); ^2^ data presented as median [interquartile range].

**Table 2 biomedicines-11-00495-t002:** Oligonucleotides used in the *GPR126* non-coding mutations ddPCR screening assay.

Oligonucleotide	Sequence	Concentration ^1^
Forward primer	5′-ATG GGA ATA TTA TAA TTC TAA CTA A-3′	0.9 µM
Reverse primer	5′-CAC CGT ATA AAT GTT CTT G-3′	0.9 µM
Wild type probe	5′-HEX-TT TGT ATG AAC ATA CAA AGA GCC TC-BHQ1-3′	0.25 µM
G/A-mutant probe	5′-FAM-TTT GTA TAA ACA TAC AAA GAG CCT C-BHQ1-3′	0.083 µM
C/T-mutant probe	5′-FAM-TTT GTA TGA ATA TAC AAA GAG CCT C-BHQ1-3′	0.083 µM
G/A + C/T-mutant probe	5′-FAM-TTT GTA TAA ATA TAC AAA GAG CCT C-BHQ1-3′	0.083 µM

^1^ Concentration in the final ddPCR mixture; *GPR126* sixth intron G/A substitution–chr. 6: 142,706,206, C/T substitution–chr. 6: 142,706,209 (assembly GRCh37); HEX, Hexachlorofluorescein; FAM, 6-Carboxyfluorescein; BHQ1, Black Hole Quencher 1.

## Data Availability

The data presented in this study are available in Appendix A.

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
