# Peer review of "Development of a Sensitive Digital Droplet PCR Screening Assay for the Detection of GPR126 Non-Coding Mutations in Bladder Cancer Urine Liquid Biopsies"

_biomedicines, 2023, doi:10.3390/biomedicines11020495_

Round 1

Reviewer 1 Report

General comment

The manuscript entitled “Development of a sensitive digital droplet Pcr screening assay for the detection of Gpr126 non-coding mutations in bladder cancer urine liquid biopsies” aims to develop a digital droplet PCR screening assay aimed to detect GRP126 mutations. Despite the interesting topic and the fair well written manuscript, the work have to be improved, especially regarding the discussion which is too rushed and lacks limitations and future perspectives. In detail:

INTRODUCTION

I would avoid to start the manuscript, as well as the abstract, with “recently”.

Regarding the role of liquid biopsy please also see: DOI: 10.3390/cancers14133272 and DOI: 10.1016/j.critrevonc.2022.103577

RESULTS

The results could be better reported. Please also check English grammar and typos.

DISCUSSION

The discussion have to be improved. Firstly, comparison with similar articles and contextualization with the current research panorama has to be implemented. Secondly, the limitations of the study are lacking and have to be reported.

Add future perspectives.

Add a proper paragraph for conclusion.

Author Response

Response to Reviewer 1 Comments

Point 1: INTRODUCTION. I would avoid to start the manuscript, as well as the abstract, with “recently”.

Response 1: Thank you for pointing this out. The first sentence of the abstract was rewritten: “Recent whole genome sequencing studies identified two novel highly recurrent mutations in the enhancer region of GPR126 in urothelial bladder cancer (UBC) tumor samples”. Due to strict limitations for words count in the abstract we still had to put the word “recent” to highligh the fact that these novel mutations were identified not in this particular article but in “recent whole genome sequencing studies”. Additionally, word “recently” was deleted from the beginning of the introduction section.

Point 2: INTRODUCTION. Regarding the role of liquid biopsy please also see: DOI: 10.3390/cancers14133272 and DOI: 10.1016/j.critrevonc.2022.103577

Response 2: Thank you for suggesting these reviews. Relevant references were added to the beginning of the introduction where the concept of liquid biopsy is briefly described. We do realize that the tradional structure of this section in articles dedicated to the liquid biopsy in UBC includes overview of routinely used instrumental / laborotary approaches for the diagnosis of UBC (cystoscopy, cytology, etc.), with their drawbacks highlighted. But we intentionally attempted to skip it, well, basically shrink it to the first paragraph. The presented manuscript describes a study which leans mote to the technical/methodological aspect of liquid biopsy (even though we assessed the diagnostic potential of the proposed screening assay). Therefore, in the introduction we attempted to emphasize on the mutational profile of UBC, to highlight why the discovery of GPR126 non-coding mutations may be so beneficial for the development of urinary tumor DNA panels, and more importantly, why does it worth a whole article dedicated to the development of a screening assay for its detection. Moreover, according to the other commentaries (for the discussion, conclusion, limitations sections) several paragraphs were added, therefore we decided to not overextend the text by adding additional information to the introduction.

Point 3: RESULTS. The results could be better reported. Please also check English grammar and typos.

Response 3: Thank you for pointing this out, correction of grammar mistakes throughout the manuscript (not only the results section) was carried out. The following additions to the results section were done:

  • Legends/footnotes of Figures 1, 2, Table 2 were extended, so that these elements were more explanatory without referring to the main text.
  • Data points were added to the box-plots.
  • Last segment of the results section was completely reworked:

“No correlation was found between tumor size and MAF determined for either of the studied mutations and for their combination (p>0.05) (Figure S2). Absence of significant associations was observed between any qualitative/quantitative urine tDNA analysis data and UBC grade, stage, history of smoking, and age (p>0.05), except moderate correlation for GPR126 MAF and age (rS=0.481, p=0.015) (Figure S3). Finally, as the developed GPR126 non-coding mutations screening assay had a relatively high false-positive MAF threshold, the impact of macrohematuria was considered. This condition may lead to abundant presence of genomic DNA from leucocytes in urine and, thus, to artificial decrease of MAF. However, in the present cohort urine samples with macrohematuria did not statistically significantly differ from the rest (30,78 [17.63–42.33] % vs. 21.12 [4.45–46.56] %, p=0.543) (Figure S4).”

  • 3 new supplementary files (19 plots in total) were added. These figures are meant to illustrate the data presented in the final part of the results section (point 3 of this response). Figure S2: Scatter plots for mutant allele fraction and tumor size; Figure S3: Comparison of mutant allele fractions and mutation detection rates with demographic and pathological characteristics of patients in the UBC group; Figure S4: Impact of macrohematuria on the mutant allelic fraction of GPR126 6th intron.

Point 4: DISCUSSION. The discussion have to be improved. Firstly, comparison with similar articles and contextualization with the current research panorama has to be implemented. Secondly, the limitations of the study are lacking and have to be reported.

Response 4: We are grateful for this comment. 1) The discussion section was modified, the comparison with similar articles, with other methodological approaches (mainly NGS), discussion the implementation of this approach for post-treatment surveillance were added (with relevant references):

In concordance with our results, the diagnostic sensitivity of the exclusive analysis of TERT promoter mutations in urinary DNA using ddPCR varies in the available publications from 46.4% to 68.3% [16–18,27,28]. In a recent study Hayashi et al. demonstrated that the inclusion of FGFR3 S249C mutation analysis increases the sensitivity of this approach for additional 8% [29]. Although ddPCR has certain advantages over NGS, such as relatively cheap and rapid sample processing, independence of bioinformatics data interpretation, its major drawback is the limitation in the number of analytical targets per DNA sample [30]. Urine liquid biopsy approaches based on NGS already achieve sensitivity of 84.0–97.4% and specificity of 97.0-100.0%, these panels include up to 50 recurrently mutated in UBC genes, such as TERT, FGFR3, TP53, PIK3CA, ERBB2, KRAS, and others [31–33]. Despite limited effectiveness in initial diagnosis and screening, ddPCR may truly excel in the field of UBC post-treatment surveillance [34]. In this case resected material may be sequenced, therefore a single personalized ddPCR mutation assay is needed to monitor the recurrence of the disease, as it was recently demonstrated by Pritchard et al. [35].

2) A new paragraph containing study limitations was added to the end of the discussion section: “This study had certain limitations. The control group consisted of healthy volunteers with no signs of urological disease, whereas in clinical application the analysis would primarily have been carried out in differential diagnosis of patients with hematuria. Mutational status of the urinary DNA was not compared to that of the paired tumor DNA samples, which may be useful to verify the absence of false mutation calls. However, robust determination of false-positive signal cutoff values (resulting in 100% specificity) at least to some extent eliminates the need to evaluate the concordance between samples. As the presented ddPCR screening assay did not allow to discriminate between GPR126 6th intron mutations, the incidence of individual substituions and their relation to pathological characteristic of tumor was not studied”.

Point 5: Add future perspectives. Add a proper paragraph for conclusion.

Response 5: Thank you for this comment. The conclusion was transferred to a separate sectoin after the discussion. Future perspectives of the GPR126 non-coding mutations analysis were indicated.

«5. Conclusions

Urine liquid biopsy is a promising non-invasive tool for the diagnosis and management of UBC. Combined analysis of non-coding mutations in GPR126 and TERT genes in the present study allowed to detect tDNA in the majority of UBC samples. Owing to its exceptionally high prevalence in this disease, mutational hotspot in GPR126 gene must become a new staple in diagnostic panels for the analysis of urinary tDNA. Moreover, being an early genetic event in the tumorigenesis, non-coding mutations in GPR126 may be an appealing target for the post-treatment monitoring of UBC recurrence. Further studies not only evaluating the diagnostic efficacy of urine liquid biopsy, but also assessing the potential economic benefit of the translation of this approach into clinical practice are awaited».

Reviewer 2 Report

This manuscript “Development of a Sensitive Digital Droplet Pcr Screening As-2 say for The Detection of Gpr126 Non-Coding Mutations in 3 Bladder Cancer Urine Liquid Biopsies” by M. Jain et al. performed a digital droplet PCR screening assay for GPR126 mutations in a single tube. I am afraid that this is merely a simple PCR screening and is not of high scientific value.

Author Response

Point 1: I am afraid that this is merely a simple PCR screening and is not of high scientific value.

Response 1: Thank you for reviewing our manuscript. The revised document containts various improvements (including 3 new supplementary figures, extension of the discussion section, study limitations, last segment of the results were reworked, and the conclusion section was extended as well). We hope that these corrections will better highlight all the work that was done, such as the assessment of the diagnostic potential of the developed screening assay in a cohort of UBC patients, comparison of its perfomance to TERT promoter mutations detection, comprehensive evaluation of various associations of mutant allele fractions and mutation detection rates with such demographic and pathologic characteristics of study participants, such as tumor grade/stage, hematuria, age, history of smoking (Figures S2, S3, S4).

Reviewer 3 Report

The Article Titled "Development of a sensitive digital droplet PCR screening assay for the detection of GPR126 non-coding mutations in bladder cancer urine liquid biopsies", authored by Mark Jain and coleagues provides data on a novel liquid biopsy technique based on digital droplet PCR that detects mutations relevant to bladder cancer in Gpr126 gene.

The study is interesting and the results support the conclusions. Even though the specificity and sensitivity of the assay is not optimal I endorse the subastantial work by the authors. Based on the presented data, it is a leap forward towards the development of a clinically relevant assay as an additional screening method of bladder cancer. 

The manuscript is well-written and presented. 

I have some minor Comments and Suggestions for Authors that are following:

-Figure 1. It would be helpful to have a self-explanatory legend with expanded information that analyse more about HEX, FAM and BHQ1 in the selected probes.

-Figure 2. As in Figure 1, I would expect that the authors would elaborate more on the legend to clearly explain the depicted data.

Author Response

Point 1: Figure 1. It would be helpful to have a self-explanatory legend with expanded information that analyse more about HEX, FAM and BHQ1 in the selected probes.

 Response 1: Thank you for pointing this out. Figure legend was expanded, following lines were added: “Probes were dual-labeled with Hexachlorofluorescein (HEX) / 6-Carboxyfluorescein (FAM) fluorescent dyes and Black Hole Quencher 1 (BHQ1) non-fluorescent chromophore”.

Additionally, lines “HEX, Hexachlorofluorescein; FAM, 6-Carboxyfluorescein; BHQ1, Black Hole Quencher 1” were added to the footnote of Table 2, as these fluorophores were mentioned there as well. This combination of fluorescent labels is, in general, standard for probes used in all available in the market digital PCR systems (not only the Bio-Rad instrument, which was used here). We hope that disclosure of these abbreviations will be sufficient for the reader to replicate this assay if necessary.

Point 2: Figure 2. As in Figure 1, I would expect that the authors would elaborate more on the legend to clearly explain the depicted data.

Response 2: This figure’s legend was expanded as well. Following lines were added: “HEX, Hexachlorofluorescein; FAM, 6-Carboxyfluorescein. Colored dots represent single droplets of emulsion carrying amplified mutated DNA (blue), wild type DNA (green), and mutated + wild type DNA (orange)”. We hope that these additions combined with the previous figure legend will allow to interpret the depicted data more clearly.

Reviewer 4 Report

Very nice, clearly written study and demonstrates the ability to detect specific alterations from urine. I have 3 main questions:

1) the the dramatic difference between the demographics of the control and UBC group in terms of age and smoking status. Within the UBC group, is there an association between detection rate and MAF with the age of diagnosis. Likewise, is there a difference between smoking history or status (current/former/never). Also, was is the association with MAF and tumor grade?

2) The authors state that the UBC cohort had urine collected "prior to any diagnostic/surgical interventions". Are they implying this was done pre-biopsy? If so, then did they analyze the urine of patients where there was no tumor present upon cystoscopy? Likewise, if this was post-cystoscopy and pre-TURBT, what was the time period and do they think irritation or inflammation resulting from the cysto effected the results. 

3) Was the urine assessed by standard cytology and how does the sensitivity/specificity compare. 

Minor comment: 

Please add figures for all analysis performed (tumor size/maf, regression models, etc...), even if results are negative. 

For box plots, please show all data points

Author Response

Response to Reviewer 4 Comments

Point 1: The dramatic difference between the demographics of the control and UBC group in terms of age and smoking status. Within the UBC group, is there an association between detection rate and MAF with the age of diagnosis. Likewise, is there a difference between smoking history or status (current/former/never). Also, was is the association with MAF and tumor grade?

Response 1: Thank you for this comment. The last paragraph of the results section was reworked:

“No correlation was found between tumor size and MAF determined for either of the studied mutations and for their combination (p>0.05) (Figure S2). Absence of significant associations was observed between any qualitative/quantitative urine tDNA analysis data and UBC grade, stage, history of smoking, and age (p>0.05), except moderate correlation for GPR126 MAF and age (rS=0.481, p=0.015) (Figure S3). Finally, as the developed GPR126 non-coding mutations screening assay had a relatively high false-positive MAF threshold, the impact of macrohematuria was considered. This condition may lead to abundant presence of genomic DNA from leucocytes in urine and, thus, to artificial decrease of MAF. However, in the present cohort urine samples with macrohematuria did not statistically significantly differ from the rest (30,78 [17.63–42.33] % vs. 21.12 [4.45–46.56] %, p=0.543) (Figure S4).”
As for smoking, unfortunately, the only available entry is “active smoker: yes/no”. So we could not compare “current/former/never”.

Point 2: The authors state that the UBC cohort had urine collected "prior to any diagnostic/surgical interventions". Are they implying this was done pre-biopsy? If so, then did they analyze the urine of patients where there was no tumor present upon cystoscopy? Likewise, if this was post-cystoscopy and pre-TURBT, what was the time period and do they think irritation or inflammation resulting from the cysto effected the results.

Response 2: Indeed sample collection was done prior to the verification of diagnosis. In fact, urine was collected from all patients with suspected bladder cancer (obviously, later a few of them were not diagnosed with UBC). However, in this particular study we included only urine samples from those who were diagnosed with UBC (n=70, as it is stated in the beginning of the materials and methods section). There was not enough urine samples from patients with non-malignant conditions to form a proper group (and we did not want to make the control group to heterogenious by adding these samples to it). However, we realize that this is a limitation of the study (this point is stated in the “study limitations” paragraph at the end of the discussion section), Therefore, we will consider forming an enire group of “non-malignant hematuria” samples for our future study, along with an expansion of patients cohorts and mutations panel.

To avoid any confusion we added following line to the text:

“In the UBC group biomaterial was obtained prior to any diagnostic / surgical interventions (therefore, the diagnosis was verified shortly after the collection of biomaterial)”.

Additionally, we added a new paragraph which describes study limitations, which touches the topic of non-malignant hematuria.

“This study had certain limitations. The control group consisted of healthy volunteers with no signs of urological disease, whereas in clinical application the analysis would primarily have been carried out in differential diagnosis of patients with hematuria. Mutational status of the urinary DNA was not compared to that of the paired tumor DNA samples, which may be useful to verify the absence of false mutation calls. However, robust determination of false-positive signal cutoff values (resulting in 100% specificity) at least to some extent eliminates the need to evaluate the concordance between samples. As the presented ddPCR screening assay did not allow to discriminate between GPR126 6th intron mutations, the incidence of individual substituions and their relation to pathological characteristic of tumor was not studied”.

Point 3: Was the urine assessed by standard cytology and how does the sensitivity/specificity compare.

Response 3: We are grateful for this comment. Unfortunately, comparison with results of urine cytology was not planned for this study, and there is no corresponding data available. However, we will consider to include it into our next study to attempt to combine cytology results with the outcome of tumor DNA analysis. So far, the presented study only aimed to introduce a new urinary biomarker along with a screening assay for its detection.

Point 4: Please add figures for all analysis performed (tumor size/maf, regression models, etc...), even if results are negative.

Response 4: Thank you for pointing this out. Supplemental figures were added for all the analysis performed. We realized that probability plots for logistic regression (with absent statistical significance) are confusing and may be difficult to interpret for the reader (especially, given that there had to be 12+ of these plots, as we tested not only the detection rates, but also MAF’s for both mutational hotspots and their combination). Thefore, we slightly adjusted the implemented statistical analysis. We used Spearman’s correlation, Fisher’s exact test, and Mann-Whitney U test to comprehensively compare all qualitative and quantitative results of tumor DNA analysis in urine with various clinical and demographic characteristcs of the study participants. This allowed us to use Box-plots and scatter plots (for correlation) which, we hope, will be easier to interpret. Corresponding supplemental figures were added.

Figure S2: Scatter plots for mutant allele fraction and tumor size; Figure S3: Comparison of mutant allele fractions and mutation detection rates with demographic and pathological characteristics of patients in the UBC group; Figure S4: Impact of macrohematuria on the mutant allelic fraction of GPR126 6th intron.

Point 5: For box plots, please show all data points

Response 5: Thank you for pointing this out. Modifications for box plots were done.

Round 2

Reviewer 1 Report

The authors improved the manuscript accordingly. No further suggestions are required.

Author Response

Point 1: The authors improved the manuscript accordingly. No further suggestions are required.

Response 1: Thank you for reviewing our manuscript.

Reviewer 2 Report

My comment is the same as before.

Author Response

Point 1: My comment is the same as before.

Response 1: Thank you for the review. Our manuscript was submitted to a Special issue dedicated to the bladder cancer. Advances in the diagnosis of this condition were among the topics of the issue. In our article we propose a novel approach for the diagnosis of bladder cancer. Discovery of TERT promoter mutations (the most frequent mutations in bladder cancer) led to the development (with subsequent publications in various high-impact journals) of assays for their analysis in liquid biopsy samples. In case of TERT promoter it was exceptionally challenging due to loop formation in that region which prevented the annealing of oligonucleotides. Today TERT promoter mutations are a staple in modern liquid biopsy diagnostic panels for this disease. Recently, two novel highly recurrent mutations in the enhancer region of GPR126 were identified. They are the second most common after TERT promoter. The design of analysis assays in this case is also quite challenging as there are 3 possible substitution combinations in the hotspot (gapped by 2 bp). We are the first to present a ready-to-use screening assay which enables analysis of these mutations as well as to demonstrate that these mutations can be reliably detected in the urine of bladder cancer patients with high frequency (while they are completely absent in the control group). We believe that GPR126 non-coding mutations will follow the path of pTERT and be actively studied world-wide. Moreover, we present the diagnostic perfomance of the assay combined with TERT mutations analysis, data regarding the associations of mutant allelic fractions with various clinical and pathological characteristics. Finally, we added results of liquid biopsy analysis of the third group of patients (cystitis), which better resembles the possible differential diagnosis in real clinical application.

Reviewer 4 Report

I appreciate the authors adding the additional figures, however it is unclear what was added that strengthened the manuscript. In fact the association between age and MAF underscores the short comings of the control cohort. The authors need to demonstrate the lack of variant DNA in a cohort of similar age. This would be of particular importance since only 25/70 UBC samples contained the variant. 

Author Response

Point 1: I appreciate the authors adding the additional figures, however it is unclear what was added that strengthened the manuscript.

Response 1: Thank you for this comment. All figures (and other corrections) were added according to the reviewers suggestions in the previous round of review. We hope that the addition of figures for all analysis performed (tumor size/maf, regression models, etc.) would allow the readers to better interpret the results.

Point 2: In fact the association between age and MAF underscores the short comings of the control cohort. The authors need to demonstrate the lack of variant DNA in a cohort of similar age. This would be of particular importance since only 25/70 UBC samples contained the variant.

Response 2: As it is written in the Figure S3 legend, MAF was subjected to the analysis only in positive samples. GPR126 mutations detection rate was not associated with age. The following addition was done to the respective sentence in the text to clarify this point:Absence of significant associations was observed between any qualitative/quantitative urine tDNA analysis data and UBC grade, stage, history of smoking, and age (p>0.05), except moderate correlation for GPR126 MAF and age (rS=0.481, p=0.015), although not for its detection rate (p>0.05) (Figure S3).”

Anyway, we cannot disagree with the reviewer that it is important to demonstrate that these mutations are not detectable in the urine of a cohort which better resembles the possible differential diagnosis situation in clinical application. Moreover, the same concern may be raised by the readers. Therefore, our team is grateful for the indication of this flaw.

As it was written in the response to the point 2 in the first review round, we collected biomaterial prior to any diagnostic/surgical interventions. Thus, in some cases after the results of histological evaluation arrived patients were excluded from the study due to absense of malignancy. In most cases these patients had various types of cystitis. These patients are few (n=22), but they are of similar age, some of them had macrohematuria, urinary retention, etc. Initially we did not want to add them to the control group to not make is heterogeneous. Thanksfully, we had their urine stored (although not analyzed) along with UBC group samples. We have consulted with the local ethics committee, these patients with cystitis had signed the same informed consent form as UBC patients (as they were enrolled basically simultaneously), our approved study protocol is in fact not stating that we should completely exclude patients from the analysis due to absense of malignancy. Therefore, we were allowed to extract the DNA and run the ddPCR in those samples. As it was expected none of them had tumor DNA detected.

Several corrections were introduced to the text regarding the addition of this third group of patients. It is now mentioned in the abstract, materials and methods, results, table 1, study limitations. The raw data of each sample from this group was added to the supplementary table S1, for data availability purposes.